# A Systematic Mapping of Research on Sustainability Dimensions at Farm-level in Pig Production

**Stefan Gunnarsson [1],\*** , **Katarina Arvidsson Segerkvist [1]** , **Torun Wallgren [1]** ,
**Helena Hansson [2]** and **Ulf Sonesson [3]**

[1]   Department of Animal Environment and Health, Swedish University of Agricultural Sciences (SLU),
     P.O.B. 234, S-53223 Skara, Sweden; katarina.segerkvist@slu.se (K.A.S.); torun.wallgren@slu.se (T.W.)
[2]   Department of Economics, Swedish University of Agricultural Sciences (SLU), P.O.B. 7013,
     S-75007 Uppsala, Sweden; helena.hansson@slu.se
[3]   RISE Research Institutes of Sweden, P.O.B. 5401, S-40229 Göteborg, Sweden; ulf.sonesson@ri.se
\*   Correspondence: stefan.gunnarsson@slu.se; Tel.: +46-511-672-16

**Abstract:** We systematically mapped the scientific literature on the sustainability of pig production at farm-level. Sustainability was considered holistically, covering its economic, environmental, and social dimensions, each consisting of a broad range of different aspects that may contradict or reinforce each other. Literature published between January 2000 and March 2020 with a geographical focus on Europe, North America, Australia, and New Zealand was included. A standard template with predefined keywords was used to summarise aspects of each sustainability dimension covered in identified papers. We found that papers analysing environmental sustainability were more frequent than papers analysing economic or social sustainability. However, there are many different aspects within each dimension of sustainability, hampering comparisons between studies. In addition, each dimension of sustainability has many sides, making it difficult to compare different studies, and different dimensions and aspects may have complex interrelations. Our systematic literature review revealed that these interrelations are not well understood and that possible trade-offs or synergies between different aspects of sustainability dimensions remain unidentified. This systematic mapping of the current literature on farm-level sustainability in pig production can support a more informed discussion on knowledge gaps and help prioritise future research at farm-level to enhance sustainability in pig production.

**Keywords:** pork; swine; environment; social; economic; animal welfare; ecology

## 1. Introduction

Pork comprises about 40% of global meat consumption and is the most commonly consumed meat [1]. The growing global human population is predicted to exceed 10 billion by 2055 [2], with a large share in eastern Asia. This, combined with a global increase in average per capita income, will increase future demand for livestock products. In 2016 there were 980 million pigs in the world [3]. Pig production provides humans with a source of animal protein, and may also constitute the main income for many farms and farming families around the world. As such, pig production contributes to several of the 17 global Sustainable Development Goals (SDG) set by the United Nation in 2016, including SDG1: zero hunger, SDG3: good health and wellbeing, and SDG8: decent work and economic growth [4]. However, pig production may also compromise other SDGs, including SDG13: climate action and SDG15: life on land. The need for the growing livestock sector to meet rising demand for animal product while also transitioning to environmentally sustainable production processes and helping to meet several social demands, such as poverty alleviation, has been

emphasised [1]. Major challenges with modern pig production include emissions of greenhouse gases, loss of biodiversity, and exhaustion of finite natural resources. For sustainable future pig production, it is of vital importance that these challenges are dealt with. Furthermore, pig production is one of most widespread farm livestock enterprises, involving many farms within Europe, Africa, Eastern Asia, and America. Thus, the sustainability of pig production has an impact locally as well as globally.

The Brundtland Commission defined sustainable development as "development that meets the needs of the present without compromising the ability of future generations to meet their own needs" [5]. Sustainable agriculture has previously been defined as management and use of agricultural ecosystems in a way that maintains biological diversity, regeneration capacity, vitality, productivity, and ability to fulfil—today and in the future—significant ecological, economic, and social functions at local, national, and global levels and does not harm the ecosystem [6]. For the livestock industry, Torp-Donner and Juga [7] defined sustainable livestock production as "production that is ecologically sound, taking into account the environment and biodiversity, ethically and economically sustainable", but pointed out that no universal definition exists. Sustainability in animal food production includes many ecological issues, but the economic dimension of sustainability is particularly important since agriculture is commonly carried out by private firms, for which revenues need to exceed costs, at least over the long term, to sustain these firms and thereby production and food supply. Although sustainability is a holistic concept, the concept needs to be split into several dimensions that can be analysed empirically, which in turn is a necessary basis for further scientific analyses on higher system levels. When discussing sustainability, one can refer to the three main dimensions; environmental, economic, and social. There are also specific aspects of animal production, such as animal welfare, impacts on public health, and ethical considerations related to animal husbandry [8]. The latter largely relate to social acceptance of production. This means that truly sustainable farm animal production requires all these aspects to be taken into account. However, the different dimensions of sustainable development and the aspects constituting each dimension may be conflicting, meaning that there are trade-offs between them or that they reinforce each other.

Environmental sustainability in pig production includes the management of harmful waste products and pollution caused by different production systems, as well as impacts caused by feed production. Emissions of greenhouse gases and excess nutrients constitute a considerable environmental challenge. Some studies have explored potential solutions to reduce the environmental impact, including methods for monitoring and assessing such wastes, along with possible bioconversion of manure to biogas and bio fertiliser [9]. Another aspect, irrespective of geographical area, is the importance of preserving biological biodiversity. The intensive use of scarce natural resources, such as land exploitation and large monocultures of grain and soy, is further a major environmental concern frequently mentioned in connection with pig production [10].

Economic sustainability in pig production, often referred to in terms of economic output, productivity, and efficiency of the production process, is often considered in association with the other dimensions of sustainable pig production. For example, prevention of zoonotic diseases, such as influenza and salmonellosis, is of major importance for public health and also has significant economic consequences [11]. Improving pig health and immunity is not only an animal welfare issue, but also affects the efficiency of the process in which production inputs can be transferred to production outputs, and thus the economic return of the process. This also applies to environmental sustainability, as reduced production efficiency often leads to inefficient resource use and hence higher impacts per unit of meat produced. Concerns about potentially large economic costs associated with improving animal welfare have been voiced by some stakeholders, for instance in a recent government inquiry in Sweden [12].

Social sustainability is a complex dimension of pig production [13]. The widespread use of antibiotics, and resulting residues and resistant bacteria in soil and food products, are a threat to human health [14,15]. The working environment may also pose a threat, exposing staff members to noise, increased risk of respiratory diseases, and injuries [16]. In addition, odour emissions from pig houses

may have a negative effect on local residents [17]. Society is increasingly demanding that agricultural production provide ecosystem services [18]. Societal acceptance of pig production is fundamental for the survival of this form of agricultural production in the future [8,19,20].

Animal welfare has been proposed as a fourth dimension of sustainable agriculture [8] and the public attitude to animal welfare is an important consideration when assessing the sustainability of animal production [19]. Broom [19] claimed that "No system can be sustainable if a substantial proportion of people finds aspects of it now, or of its consequences in the future, morally unacceptable". In 2016, the United Nations Committee on World Food Security (UN-CFS) included improving animal welfare as a goal in its draft recommendations on sustainable agricultural development [21]. This places animal welfare within the same concept as the three classical dimensions (economic, environmental, and social sustainability), even though Buller, et al. [22] argued that animal welfare is yet to be a fully integrated component of sustainability. Animal welfare is often considered in association with other dimensions of sustainability. For example, improved animal health and reduced use of antibiotics are not only important for pig welfare, but also for economic reasons (reduced costs, increased productivity, and product quality), social reasons (improved public health, public acceptance of production) and environmental reasons (increased resource efficiency, reduced drug residues, and resistant bacteria in soil) [23]. Some authors have tried to incorporate animal welfare in life cycle analyses (LCA) of animal production [24]. Genetic selection for increased productivity has been proposed, but without taking animal welfare into further consideration, alongside a growing concern that intensive farming may threaten animal welfare [25]. Other studies have discussed potential economic consequences for farmers working to enhance animal welfare [26,27].

In order to develop sustainable pig production that meets both the needs and demands of the future, a useful starting point would be to map current knowledge and based on this, identify major knowledge gaps. Accordingly, the aim of this study was to systematically map the scientific literature on sustainability in pig production, covering a wide area of aspects within the three main dimensions of sustainable development, including e.g., feed, genetics, animal health, production systems, profitability, economic efficiency, labour, employment, and quality in primary production. Systematic mapping aims to catalogue available papers related to a specific topic in order to identify knowledge and knowledge gaps, as opposed to a systematic review, which aims to answer a specific question [28]. We focused on studies conducted in Europe, North America, Australia, and New Zealand to include production systems based on similar economic contexts. To the best of our knowledge, no previous systematic mapping has been performed of the scientific literature related to sustainable pig production. Our intention was to provide important insights into the current state of knowledge, identify areas for future research, and provide a basis for prioritising research areas.

## 2. Materials and Methods

### 2.1. Definition of Fundamental Concepts

Environmental sustainability is a natural science-based dimension and, in principle, concerns the total impacts on ecosystems caused by human activities. The most commonly used tool for quantification of environmental impacts is a life cycle assessment (LCA). Within LCA, flows of resources (energy, land, water etc.) into the production system and emissions from the system are quantified and potential impacts described. The environmental impact caused by the emissions, at all spatial scales, should be included, as should the perspective of resource availability locally, regionally, and globally. The scope of life cycle assessments (LCAs) of environmental sustainability should be the function of the production, not the organisational boundaries. Thus all flows to and from the production system need to be considered, regardless of economic ownership [29]. There are other assessment methods besides LCA, most of which are similar in terms of their basic principles, but may use different approaches, in particular in the description of impacts. An example is the ecological footprint approach, where all emissions and resource use are transformed into "hypothetical area

used", thus creating a common unit for the environmental impact of individual products [30]. In the present study, the concept of environmental sustainability was taken to include topics such as: land use, biodiversity, greenhouse gas emissions, eutrophication, pollution, and other environmental hazards.

Concerning economic sustainability, this dimension of sustainable development has generally come to include aspects such as job creation and income generation to sustain the population. However, the scientific literature is not clear on how best to measure economic sustainability [31]. Within sustainability accounting, the Global Reporting Initiative (GRI) reporting standards [32] provide guidance on which aspects to consider, and include economic indicators such as costs, revenues, profit, and investments. The economic dimension of sustainable development basically implies a focus on growth of the economic system and on maintaining the capital invested in firms. In this respect, an interesting distinction can be made between weak and strong sustainability [33,34], and thus natural and economic capital. Weak sustainability is about maintaining the sum of those two types of capital together, whereas strong sustainability is about maintaining each type separately [34].

Sustainability in an economic perspective can also focus on sustainable use of natural resources within a defined economic system, basically meaning that sustainability is achieved when the economic activity is not undertaken at the cost of natural resources. The economic concept of negative externalities is useful in understanding and fully capturing all costs associated with production, i.e., not only costs incurred by the producer, but also societal costs [34]. In the present analysis, we assessed whether the selected literature considered economic indicators such as profit, economic efficiency, cost, and returns on capital in sustainability assessments.

*Social sustainability* is less well-defined and mostly a neglected dimension of sustainability [13]. It may be defined as the ability of a community to develop processes and structures that meet the needs of community members and, furthermore, support the ability of future generations to maintain a healthy community. However, there is no universal definition of social sustainability, and it has hence been defined in different ways and in relation to the other two dimensions [35]. One proposed broad definition that has been suggested is that social sustainability is a life-enhancing condition within communities, and a process within communities that can achieve that condition [36,37]. In the present study, the concept of social sustainability was taken to include topics such as social equity, livelihood, health equity, community development, labour rights, and community resilience.

## 2.2. A Systematic Mapping Approach

Systematic mapping, also known as evidence gap mapping, was used to evaluate the current literature on farm-level sustainability in pig production. Systematic mapping is a transparent, robust, and repeatable method for identifying and collecting relevant literature on a research question [38].

## 2.3. Search of Literature

Comprehensive searches of several information sources were carried out in an attempt to obtain an unbiased sample of published literature. The searches were conducted in the end of March 2020. The following online literature databases were used to identify relevant literature:

- Scopus;
- Web of Science Core Collection;
- CABI: Cab Abstracts.

The time span for the searches was set to papers published from 1 January 2000 to 20 March 2020. In order to find relevant papers dealing with pigs, the following search terms were used: pig OR pigs or piglet* OR sow OR sows OR swine OR pork. These were combined with search terms for the three dimensions of sustainability (Table 1), which were defined through an iterative process by all co-authors, in collaboration with a university librarian. The authors have expertise in different areas and were able to suggest possible search terms and list indicators of sustainability typically used in each area. In Scopus, the search was made within the search field "Title, Abstract, Keywords" and

in Web of Science Core Collection and CABI: Cab Abstracts, the searches were made within "Topic", which includes title, abstract, and keywords.

**Table 1.** Library search terms for environmental, economic, and social sustainability (will be moved).

| Sustainability Dimensions | Search Terms [1] |
|---|---|
| Environment | ("environmental impact assessment" OR (environment* NEAR/2 assessment) OR (environment* NEAR/2 impact) OR (environment* NEAR/2 protection) OR (climate NEAR/1 change*) OR biodiversity OR ecosystem* OR pollution OR deforestation OR eutrophication OR (habitat NEAR/2 destruction) OR (land NEAR/2 degradation) OR (ozone NEAR/2 depletion) OR "acid deposition" OR (odour NEAR/2 emission) OR "air quality" OR "biochemical oxygen demand*" OR "chemical oxygen demand*" OR (nitrogen NEAR/2 balance) OR (nitrogen NEAR/2 cycle) OR (carbon NEAR/2 cycle) OR eco-toxicity OR "carbon footprint" OR LCA OR "life cycle assessment") |
| Economic | (agricultur* NEAR/2 development) OR (agricultur* NEAR/2 production) OR (farm* NEAR/2 comparison*) OR (farm NEAR/2 entrant*) OR (farm NEAR/2 result*) OR (farm NEAR/2 development) OR production OR diversification OR intensification OR "technical efficiency" OR "economic efficiency" OR "eco-efficiency" OR profit OR econom* OR return OR "economic viability" OR "economic performance") |
| Social | (attitude* NEAR/2 work) OR labour OR labor OR (quality NEAR/2 life) OR "living condition*" OR "rural welfare" OR (work* NEAR/2 condition*) OR "rural development" OR "social welfare" OR "social security" OR "social service*" OR "social equity" OR (health NEAR/2 service*) OR "social status" OR (women NEAR/2 status) OR "equal right*" OR equality OR (rural NEAR/2 employment) OR livability OR "health equity" OR "labour rights" OR "labor rights" OR "social justice" OR "social capital" OR (community NEAR/2 development) OR (community NEAR/2 resilience) |

[1] For the searches in Web of Science Core Collection and CABI: CAB Abstracts, the Boolean operator NEAR was used and for the search in Scopus, the Boolean operator W was used.

The results of the searches were imported into EndNote X8™. A separate library was made for each search in the different databases. When searching was complete, all the libraries were incorporated into one new library, and the number of references found was recorded. Any duplicates were removed using the automatic function in the EndNote X8™ software. The retrieved library was then manually searched for references relevant to the topic. Only full-length, trial-based papers were included, i.e., literature reviews, book chapters, conference papers, and organisation reports were excluded. Further, the full-length papers had to be written in English to be included. In addition to the articles addressing various dimensions of sustainability, a geographical limitation was also set, which meant that only studies conducted in Europe, North America, Australia, and New Zealand were included, to obtain studies conducted on production systems in similar economic contexts. Exclusion due to publication type, language, or geographical origin was performed manually, and was not set up in the searches.

In the next phase data were extracted from the abstracts of the studies included, in order to describe important aspects of the three sustainability dimensions, using a template. In the template, keywords used in each study to describe sustainability were defined and categorised. The full text of selected papers that listed aspects of the dimensions of sustainability in the abstract were analysed in order to assess the actual scientific content of all three dimensions, i.e., that all dimensions were empirically studied.

## 3. Results

The literature searches resulted in a total of 589 hits, of which 362 hits (61%) originated from CABI: Cab Abstracts, 106 hits (18%) originated from Scopus, and 121 (21%) hits originated from Web of Science Core Collection. Review papers, papers not written in English, and papers not referring to countries within the scope of this study were removed, leaving 36 papers that were included in the analysis (Appendix A). No reported studies conducted in Australia or New Zealand were found.

The 36 selected papers were screened to ensure that they covered all three sustainability areas and belonged to the target geographical area.

On assessing the abstracts of the 36 papers, we found that only seven included key words that were associated with all three dimensions of sustainability (environmental, economic, and social). Seven abstracts included two dimensions (economic and social, environmental and social, and economic and environmental in one, three, and three papers, respectively) and one paper involved one of the dimensions (environmental). This left us with a total of 15 relevant papers (Figure 1), of which nine papers described empirical experiments. Table 2 summarises characteristics of the selected 15 papers with respect to (i) what indicators of each sustainability dimension have been used, (ii) under what wider label indicators under each sustainability dimension can be grouped to facilitate comparison, and (iii) number of papers that have investigated sustainability under each label.

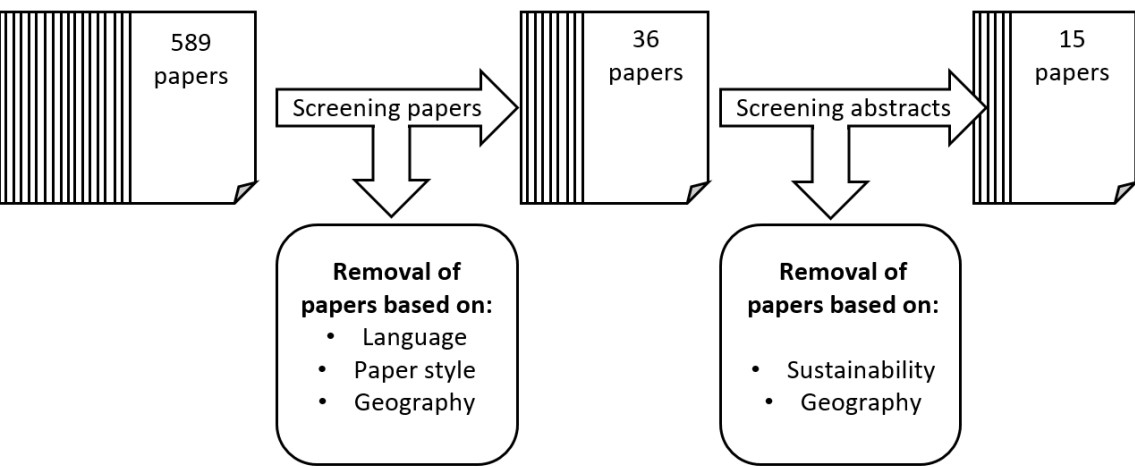

**Figure 1.** A schematic picture of the selection process of papers included in the systematic mapping.

**Table 2.** Indicators of environmental, economic, and social sustainability and research methods used in the selected papers (n = 15), and number of times each summarised category was used in these papers.

| Economic Dimension | Summarised Economic | Number of Papers | Environmental Dimension | Summarised Environmental | Number of Papers [1] | Social Dimension | Summarised Social | Number of Papers |
|---|---|---|---|---|---|---|---|---|
| Economic viability Net farm income Income generation Producer surplus | Firm economic viability | 4 | Climate change Acidification Land occupation Eutrophication | LCA based | 5 | Animal welfare Animal health Use of antibiotics Pig mortality rate | Animal health and welfare | 8 |
| Financial stability Net present value Internal rate of return | Financial situation and returns to capital | 3 | Soil quality, erosion, and C accumulation Soil quality, nutrients Biodiversity Maintenance of ground water | Local ecosystem services | 4 | Breeding programs | | |
| Transferability Generating capacity Market conformity Consumer surplus | Market adaptation and consumer perspectives | 4 | Ammonia emissions Nitrogen losses from soil and manure Odour control and emissions reductions | Local emissions | 5 | Working conditions Occupational health Job creation Local income Employment Meat safety | Employment and working conditions | 6 |
| Costs | Costs | 3 | Nitrogen Phosphorus Energy Transport | Global resources | 6 | Food security | Food security | 4 |
| Efficiency Labour productivity Productivity | Efficiency and productivity | 4 | | | | Changes in agricultural structure Political and social possibilities to control production | Legislation and regulation | 2 |
| Dependence on subsidies Governmental payments | Subsidies | 3 | | | | Stakeholder perceptions Social acceptability Cultural acceptability Landscape aesthetics Appreciation of the region Odour | Societal acceptance | 8 |
| Production management | Management | 1 | | | | sLCA | sLCA | 1 |

[1] Note: Papers typically considered more than one aspect within each dimension of sustainable development.

## 4. Discussion

Systematic mapping is a useful approach for obtaining an overview of current scientific literature. In systematic mapping, searches and inclusion/exclusion process are conducted with the same comprehensive method as for a full systematic review, but the process does not extend to critical appraisal or data synthesis [38–40]. The results obtained through systematic mapping enable identification of relevant knowledge gaps in the context of stakeholder knowledge and opinions. This information can then be used to establish a tentative agenda for high-priority areas for future research involving stakeholders—in the present case with the primary focus on pig production in high-income countries. The data extracted from the dataset included in the present analysis described important aspects of the different dimensions of sustainability dealt with in the studies. A major benefit of systematic mapping is that vast and potentially diverse research areas can be investigated (mapped) in a comprehensive way, providing a useful overview of the area and priorities and the main focus applied in earlier research. This provides useful information on how to prioritise research in the future by indicating knowledge gaps that can be evaluated by researchers and stakeholders. In contrast, a systematic review requires substantial analysis of a minor and thoroughly specified topic, where the results from the selected papers are statistically evaluated and synthesised [28]. For the purposes of the present study, a systematic mapping approach was considered more useful.

With our search method, we identified papers where the words in the search string were included in the title, abstract, or keywords. However, the papers produced by the searches did not necessarily have sustainability as part of the aim. Furthermore, the aims stated were often not those we were initially interested in, or were outside the scope of the analysis. We excluded studies without clearly stated aims about investigating sustainability, but these may well have covered aspects of sustainability that were relevant to our study.

Most papers identified were not in areas that were relevant to our topic, i.e., farm-level sustainability in pig production. Although our searches contained search terms for all three dimensions of sustainability, only a limited proportion of papers (15 out of 36) were initially identified as actually dealing with all dimensions (environmental, economic, and social). Further investigation of the 15 papers revealed that only five of them actually fell within the defined scope, leaving us with an even smaller proportion (five out of the 36 relevant papers). In the rejected papers, sustainability was not part of the aim or the content of the paper did not meet our set criteria, including all three pillars of sustainability. Further investigation revealed that only five papers fully fit the set criteria. Of the five final remaining papers that met our inclusion criteria, two were methodology papers describing methods and procedures for evaluating sustainability [41,42] and were part of a larger project where the actual results were published in other papers. The outcomes of our search method were therefore rather limited, and it was rare for all three dimensions of sustainability to be actually investigated within the same study.

Of the few papers that covered all dimensions of sustainability, some described a single scientific study and reported the results on the different sustainability dimensions in separate papers, e.g., works performed by Bonneau and co-workers [41–44]. This indicated that we might have retrieved an incomplete set and missed relevant papers by only including papers concerning all three dimensions. Therefore, we performed a further analysis of papers covering one or two of the three sustainability dimensions regarding the scope in order to analyse the relevance. However, the lack of papers discussing and evaluating all three sustainability dimensions in relation to each other might lead to loss of significant information of importance for the development of sustainable pig production. There may also be other papers that covered all three dimensions of sustainability, but which did not appear among the hits because they did not include the exact words used in our search strings. However, to ensure achieving full coverage of the dimensions, the search string was developed through an iterative discussion within the project group and with a qualified university librarian. This made the search strings rather comprehensive, so we believe that we included a majority of studies dealing with sustainability within pig production at farm-level.

Another consideration is that using our search strings resulted mainly in papers that were considered to be outside the scope of our analysis. The comprehensive search strings included all papers containing one or more words from each group of words connected to each sustainability dimension in our search string according to the string set-up. The identification of 'false' papers (i.e., containing key words, but not covering the topics of interest) could have been minimised with a more specific search string, either containing fewer words or more specific searches, for instance for matches only within the title, instead of in title, abstract, and key words. However, such an approach would have increased the risk of missing relevant papers. The wide range of hits due to the comprehensive search string was instead manually assessed to identify relevant papers from the abstract, so the risk of deleting relevant papers was low.

Initially, we did not include animal welfare as a criterion in social sustainability, but we added it to this dimension during the reading process, as it is closely linked to society's view of animal production. However, it is not straightforward to decide which dimension of sustainable development animal welfare fits best into, or if it should be considered a dimension on its own. In particular, animal welfare can be considered related to all parts of sustainable development. This can be clarified through some examples: (i) Grazing animals are allowed to express their natural behaviour and should thus experience higher levels of animal welfare. At the same time, grazing animals often help to improve biodiversity, which is part of the environmental dimension of sustainable development. (ii) Animal health is part of animal welfare and having healthier animals reduces spending on veterinary treatments, discarded agricultural products, and labour requirements, which is part of both the economic and environmental dimensions of sustainable development. (iii) Improved animal welfare is likely to improve society's acceptance of animal production, which is part of the social dimension of sustainable development. Animal welfare, although it may be defined in different ways [45], was then included as part of social sustainability. It should be noted that other aspects of sustainable development perspectives may be interrelated in similar ways. However, it is clear from our systematic mapping of the current literature that those interrelations are not well-understood and that possible trade-offs or synergies between different aspects have not yet been identified. This is an important area for future research and highlights the importance of all aspects of sustainability being discussed and evaluated within the same paper.

However, improving pig health and immunity is not only an animal welfare issue, but also affects the efficiency of the process by which production inputs are converted into production outputs, and thus the economic return of the process. This also applies to environmental sustainability, as reduced production efficiency often leads to inefficient resource use, and hence higher impacts per unit of meat produced. However, concerns about potentially large economic costs associated with improving animal welfare have been voiced by stakeholders, for instance in a recent government inquiry in Sweden [12].

Few papers in the dataset covered all three dimensions of sustainability together and none of the papers assessed whether or how different sustainability dimensions can counteract or reinforce each other. It is difficult to estimate and take full advantage of the aspects of environmental, economic, and social sustainability within a particular production system at the same time. The various sustainability aspects therefore need to be weighed against each other in order to maximise the overall sustainability at farm-level, or even in a broader societal context. According to Hansen (1996), sustainability can be defined at different systems levels—farm, region, and global. At the base level, the farm, production must be sustained environmentally and economically over a long time. The higher levels, regional and global, add more requirements that need to be met, but they build on sustainable farms. Hence, farm-level sustainability is crucial for larger-scale sustainability [46]. This emphasises the need for studies investigating several aspects at the same time, and their relations to each other. Furthermore, it is probably difficult to maximise environmental, economic, and social sustainability within the same production system. Therefore, various aspects of sustainability need to be balanced in order to maximise the overall sustainability within a farm, or even in a broader societal context.

Initially, we aimed to identify papers presenting research conducted in Europe, North America, Australia, and New Zealand. These geographical areas were included due to expected similarities between production systems, enabling comparisons between studies. However, we did not identify any papers from Australia and New Zealand, probably due to the minor importance of pig production in these countries.

Regarding the definitions of the sustainability criteria, a clear challenge was the very wide definitions of the three dimensions of sustainability that are used in the scientific literature. For example, the term environmental sustainability was used to describe studies ranging from energy use in a single process line to full-scale cradle-to-gate LCA studies. There was similar, or even larger, variety in the definitions of social and economic sustainability. In addition, there are many different aspects within each dimension, which complicated comparisons between studies since there may have been differences in the approach used to investigate, e.g., environmental sustainability may differ radically between studies. Future research should focus on developing a taxonomy for conceptualising various aspects of the three dimensions of sustainability, which could enable comparison between studies.

In most of the papers in the dataset there was a clear focus on one dimension of sustainability, which was quantified, while the other dimensions were discussed more briefly, mainly to put the main results in context. As concluded by authors such as Bonneau and co-workers [42], regarding sustainability, we found that few previous studies have investigated all three dimensions of sustainability in pig production, with more papers covering just one dimension. We found more papers dealing with environmental or social sustainability than economic sustainability. These findings can be used for prioritising future research related to the sustainability of pig production, by comparing current knowledge against identified needs for future knowledge. In the future, an interdisciplinary study probably should be performed in order to develop a conceptual framework for a sustainability performance assessment of pig production at farm and territorial level, where all three dimensions of sustainability are assessed, including potential synergies and conflicts between sustainability goals and targets.

## 5. Conclusions

We identified the scientific papers on sustainability in pig production, including all dimensions of sustainable development. In the retrieved literature, we found few studies that included all three dimensions of sustainability simultaneously, but papers covering one of the three dimension of sustainability were more common. Papers that were dealing with environmental sustainability were more frequent than papers analysing economic or social sustainability. Our findings can be used for prioritizing future research to understand the interplay between different aspects and how this can affect the development of sustainable pig production.

**Author Contributions:** conceptualisation, S.G., H.H., U.S., and K.A.S.; investigation, K.A.S. and T.W.; data curation, T.W.; formal analysis, K.A.S. and T.W.; writing—original draft preparation, T.W. and S.G.; review and editing, H.H., U.S., K.A.S., and S.G.; funding acquisition, S.G. All authors have read and agreed to the published version of the manuscript.

**Funding:** The Swedish Research Council Formas, grant number 2017-02017, funded this research.

**Acknowledgments:** We thank Mattias Lennartsson, librarian at the Swedish University of Agricultural Sciences, for valuable help during the process of developing the search strings.

**Conflicts of Interest:** The authors declare no conflict of interest. The funders had no role in the design of the study; in the collection, analyses, or interpretation of data; in the writing of the manuscript; or in the decision to publish the results.

## Appendix A

The 36 papers included in systematic mapping that included all three sustainability dimensions (economic, environmental, and social).

I.  Ashwood, B.M. Rural residents for responsible agriculture: hog CAFOs and democratic action in Illinois. *J. Rural Soc. Sci.* **2013**, *28*, 76–88.

II.  Baxter, E.M.; Jarvis, S.; Sherwood, L.; Farish, M.; Roehe, R.; Lawrence, A.B.; Edwards, S.A. Genetic and environmental effects on piglet survival and maternal behaviour of the farrowing sow. *Appl. Anim. Behav. Sci.* **2011**, *130*, 28–41.

III.  Blanes-Vidal, V.; Hansen, M.N.; Adamsen, A.P.S.; Feilberg, A.; Petersen, S.O.; Jensen, B.B. Characterization of odor released during handling of swine slurry: Part II. Effect of production type, storage and physicochemical characteristics of the slurry. *Atmos. Environ.* **2009**, *43*, 3006–3014.

IV.  Bonneau, M.; De Greef, K.; Brinkman, D.; Cinar, M.U.; Dourmad, J.Y.; Edge, H.L.; Fàbrega, E.; Gonzàlez, J.; Houwers, H.W. J.; Hviid, M.; Ilari-Antoine, E.; Klauke, T.N.; Phatsara, C.; Rydhmer, L.; Van Der Oever, B.; Zimmer, C.; Edwards, S.A. Evaluation of the sustainability of contrasted pig farming systems: The procedure, the evaluated systems and the evaluation tools. *Animal*, **2014**, *8*, 2011–2015.

V.  Bonneau, M.; Klauke, T.N.; Gonzàlez, J.; Rydhmer, L.; Ilari-Antoine, E.; Dourmad, J.Y.; De Greef, K.; Houwers, H.W.J.; Cinar, M.U.; Fàbrega, E.; Zimmer, C.; Hviid, M.; Van Der Oever, B.; Edwards, S.A. Evaluation of the sustainability of contrasted pig farming systems: Integrated evaluation. *Animal*, **2014**, *8*, 2058–2068.

VI.  Bordeaux, C.; Grossman, J.; White, J.; Osmond, D.; Poore, M.; Pietrosemoli, S. Effects of rotational infrastructure within pasture-raised pig operations on ground cover, soil nutrient distribution, and bulk density. *J. Soil Water Conserv.* **2014**, *69*, 120–130.

VII.  Bottoms, K.; Poljak, Z.; Friendship, R.; Deardon, R.; Alsop, J.; Dewey, C. An assessment of external biosecurity on Southern Ontario swine farms and its application to surveillance on a geographic level. *Can. J. Vet. Res.* **2013**, *77*, 241–253.

VIII.  Brambilla, M.; Navarotto, P. Sensorial analysis of pig barns odour emissions. *Chem. Eng. Trans.* **2010**, *23*, 243–248

IX.  Cairns, K.; McPhail, D.; Chevrier, C.; Bucklaschuk, J. The family behind the farm: race and the affective geographies of Manitoba pork production. *Antipode* **2015**, *47*, 1184–1202.

X.  Camara, E.E.G.da; Duarte, E.A.; Ferreira, L. Overall assessment of environmental impacts of animal production in Portugal. *Anais do Instituto Superior de Agronomia*, **2000**, *48*, 9–40.

XI.  Dios, M.; Souto, J.A.; Casares, J.J. Emissions inventory analysis for an urban (industrial)-rural (agricultural) environment. *WIT Trans. Ecol. Environ.* **2010**, 383–392.

XII.  Dolman, M.A.; Vrolijk, H.C.J.; de Boer, I.J.M. Exploring variation in economic, environmental and societal performance among Dutch fattening pig farms. *Livest. Sci.* **2012**, *149*, 143–154.

XIII.  Glenna, L.L.; Mitev, G.V. Global neo-liberalism, global ecological modernization, and a swine CAFO in rural Bulgaria. *J. Rural Stud.* **2009**, *25*, 289–298.

XIV.  Kiley-Worthington, M. Ecological agriculture. Integrating low input, high productive farming with wildlife conservation. Results from the Experimental Farm La Combe, Drome France. *Open J. Ecol.* **2014**, *4*, 744–763.

XV.  Mann, S.; Kogl, H. On the acceptance of animal production in rural communities. *Land Use Policy* **2003**, *20*, 243–252.

XVI.  Mirabelli, M.C.; Wing, S.; Marshall, S.W.; Wilcosky, T.C. Race, poverty, and potential exposure of middle-school students to air emissions from confined swine feeding operations. *Environ. Health Perspectives* **2006**, *114*, 591–596

XVII.  Mitchell, B. Participatory partnerships: Engaging and empowering to enhance environmental management and quality of life? *Soc. Indic. Res.* **2005**, 123–144.

XVIII.  Mueller, S. Manure's allure: variation of the financial, environmental, and economic benefits from combined heat and power systems integrated with anaerobic digesters at hog farms across geographic and economic regions. *Renew. Energy* **2007**, *32*, 248–256.

XIX.     Nainggolan, D.; Termansen, M.; Reed, M.S.; Cebollero, E.D.; Hubacek, K. Farmer typology, future scenarios and the implications for ecosystem service provision: a case study from south-eastern Spain. *Reg. Environ. Change* **2013**, *13*, 601–614.

XX.      Nordborg, M.; Sasu-Boakye, Y.; Cederberg, C.; Berndes, G. Challenges in developing regionalized characterization factors in land use impact assessment: impacts on ecosystem services in case studies of animal protein production in Sweden. *Int. J. of Life Cycle Ass.* **2017**, *22*, 328–345.

XXI.     Novek, J. Intensive livestock operations, disembedding, and community polarization in Manitoba. *Soc. Nat. Resour.* **2003**, 567–581.

XXII.    Petit, J.; van der Werf, H.M.G. 2003. Perception of the environmental impacts of current and alternative modes of pig production by stakeholder groups. *J. of Environ. Manage.* **2003**, *68*, 377–386.

XXIII.   Philippe, F.X.; Laitat, M.; Nicks, B.; Cabaraux, J.F. 2012. Ammonia and greenhouse gas emissions during the fattening of pigs kept on two types of straw floor. *Agr. Ecosyst. Environ.* **2012**, 45–53.

XXIV.   Röös, E.; Patel, M.; Spangberg, J.; Carlsson, G.; Rydhmer, L. Limiting livestock production to pasture and by-products in a search for sustainable diets. *Food Policy* **2016**, *58*, 1–13.

XXV.    Röös, E.; Sundberg, C.; Tidåker, P.; Strid, I.; Hansson, P.A. Can carbon footprint serve as an indicator of the environmental impact of meat production? *Ecol. Indicators* **2013**, *24*, 573–581.

XXVI.   Sasu-Boakye, Y.; Cederberg, C.; Wirsenius, S. Localising livestock protein feed production and the impact on land use and greenhouse gas emissions. *Animal*, **2014**, *8*, 1339–1348.

XXVII.  Sato, P.; Hotzel, M.J.; von Keyserlingk, M.A.G. American Citizens' Views of an Ideal Pig Farm. *Animals*, **2017**, *7*, 64.

XXVIII. Savard, M. Modelling risk, trade, agricultural and environmental policies to assess trade-offs between water quality and welfare in the hog industry. *Ecol. Model.* **2000**, *125*, 51–66.

XXIX.   Scotford, I.M.; Williams, A.G. 2001. Practicalities, costs and effectiveness of a floating plastic cover to reduce ammonia emissions from a pig slurry lagoon. *J. Agr. Eng. Res.* **2001**, *80*, 273–281.

XXX.    Shen, H.; Henkelmann, B.; Rambeck, W.A.; Mayer, R.; Wehr, U.; Schramm, K.W. The predictive power of the elimination of dioxin-like pollutants from pigs: an in vivo study. *Environ. Int.* **2012**, *38*, 73–78.

XXXI.   Sonesson, U.G.; Lorentzon, K.; Andersson, A.; Barr, U.K.; Bertilsson, J.; Borch, E.; Brunius, C.; Emanuelsson, M.; Göransson, L.; Gunnarsson, S.; Hamberg, L.; Hessle, A.; Kumm, K.I.; Lundh, Å.; Nielsen, T.; Östergren, K.; Salomon, E.; Sindhöj, E.; Stenberg, B.; Stenberg, M.; Sundberg, M.; Wall, H. 2016. Paths to a sustainable food sector: integrated design and LCA of future food supply chains: the case of pork production in Sweden. *Int. J. of Life Cycle Ass.* **2016**, *21*, 664–676.

XXXII.  Tajik, M.; Minkler, M. Environmental justice research and action: A case study in political economy and community-academic collaboration. *Int. Q. Community Health Educ.* **2006**, 213–231.

XXXIII. Tajik, M.; Muhammad, N.; Lowman, A.; Thu, K.; Wing, S.; Grant, G.R. Impact of odor from industrial hog operations on daily living activities. *New Solutions* **2008**, *18*, 193–205.

XXXIV. Tyndall, J. Characterizing pork producer demand for shelterbelts to mitigate odor: an Iowa case study. Agrofor. Syst. **2009**, *77*, 205–221.

XXXV.  Valino, V.; Perdigones, A.; Iglesias, A.; Garcia, J.L. Effect of temperature increase on cooling systems in livestock farms. *Clim. Res*. **2010**, *44*, 107–114.

XXXVI. Vaneeckhaute, C.; Styles, D.; Prade, T.; Adams, P.; Thelin, G.; Rodhe, L.; Gunnarsson, I.; D'Hertefeldt, T. Closing nutrient loops through decentralized anaerobic digestion of organic residues in agricultural regions: A multi-dimensional sustainability assessment. *Resour. Conserv. Recycl.* **2018**, *136*, 110–117.

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
