# Peer review of "A Systematic Mapping of Research on Sustainability Dimensions at Farm-level in Pig Production"

_sustainability, doi:10.3390/su12114352_

Round 1

Reviewer 1 Report

Very clearly written. Search details enable repetition - a repeat may prove useful in the future to evaluate whether the gaps are being filled.

Author Response

Revsion has been performed according to comments from reviewer 2 and editor. 

Reviewer 2 Report

At the beginning I would like to thank the author for preparing such a publication. The concept of sustainable is gaining importance in many areas of life, including animal production.

In my opinion the work is properly prepared. Introduction, material and methods, results and discussion are described in detail. The work covers a very long period of time. The manuscript addresses many important aspects related to pig production and its impact on various spheres.

Only where I find minor shortcomings are References. Once again, review the References carefully and correct them as recommended by the Journal.

Author Response

The references list in the appendix and in manuscript reference list have been revised been according to recommendation.
